# Path Analysis of Causal Factors Influencing Marine Traffic Accident via Structural Equation Numerical Modeling

**Shenping Hu [1,\*] , Zhuang Li [1], Yongtao Xi [1], Xunyu Gu [2] and Xinxin Zhang [1]**

[1]  Merchant Marine College, Shanghai Maritime University, Shanghai 201306, China;
    shmtu_lizhuang@163.com (Z.L.); xiyt@shmtu.edu.cn (Y.X.); xxzhang@shmtu.edu.cn (X.Z.)
[2]  Ship Administration, Pudong Maritime Safety Administration, Shanghai 200135, China; jsntqdgxy@163.com
\*  Correspondence: sphu@shmtu.edu.cn; Tel.: +86-21-3828-2947

**Abstract:** Many causal factors to marine traffic accidents (MTAs) influence each other and have associated effects. It is necessary to quantify the correlation path mode of these factors to improve accident prevention measures and their effects. In the application of human factors to accident mechanisms, the complex structural chains on causes to MTA systems were analyzed by combining the human failure analysis and classification system (HFACS) with theoretical structural equation modeling (SEM). First, the accident causation model was established as a human error analysis classification in sight of a MTA, and the constituent elements of the causes of the accident were conducted. Second, a hypothetical model of human factors classification was proposed by applying the practice of the structural model. Third, with the data resources from ship accident cases, this hypothetical model was discussed and simulated, and as a result, the relationship path dependency mode between the latent independent variable of the accident was quantitatively analyzed based on the observed dependent variable of human behavior. Application examples show that relationships in the HFACS are verified and in line with the path developing mode, and resource management factors have a pronounced influence and a strong relevance to the causal chain of the accidents. Appropriate algorithms for the theoretical model can be used to numerically understand the safety performance of marine traffic systems under different parameters through mathematical analysis. Hierarchical assumptions in the HFACS model are quantitatively verified.

**Keywords:** maritime traffic; marine accident; accident causation theory; human factor; structural equation modeling; HFACS; path dependency

## 1. Introduction

Marine traffic safety is an important component of economics and trade between different countries. The volume of ship transportation has, over time, become an important measurement of a country's economic development [1]. With the growth of China's national economy, the shipping industry has developed rapidly and the scale of transportation has expanded. With that growth, marine traffic accidents (MTAs) have consistently highlighted the importance of life safety, property safety, and environment protection. Therefore, as a basic issue of safety research, the symptomatic problems of MTAs receive much interest from experts [2].

In order to reduce the incidence of MTAs, many experts have conducted research on the causes of MTAs. Marine traffic is a complex system that includes people, ships, and environmental management. In the past, people focused on improving the safety of ships and equipment. Due to the continuous development of technology, the safety of ships and equipment has reached a very high level. Safety

experts and scientists agree that the role and status of human factors and management factors in accidents have been proven. Thus, at present, many scholars believe that the root cause of accidents is management factors, i.e., the direct cause of accidents is the unsafe acts of personnel [3,4].

The development of accident causation theory shows that most accidents are not caused by a single elementary event, but by a series of factors interacting with each other. Therefore, it is necessary to study the relationship between the different causes of MTAs, in order to help decision-makers better understand the accident and thus fundamentally reduce the occurrence of such accidents. Analyses of the causes of MTA and research on the interrelationship of such causes are being continuously developed. The complexity of the cause of the accident system has been established, and the chain model associated with the cause of the accident has basically been consistent [4,5].

However, it remains a difficult to explore the association pattern and intensity of the generic causal chain quantitatively. It is possible to use new algorithms to study the interactions and influence paths of the causes of accidents. In particular, the analysis of the causal chain path of big data can help us understand the characterization mechanisms of accidents and provide scientific diagnosis of how those accidents occurred. To quantitatively analyze the relationships between the causes of MTAs and clarify the causal mechanism of human factors in an accident and analyze the logical cause of the accident, this paper will combine accident data, using the Structural equation model (SEM) method to analyze the complex relationship between the causal structures of MTA system.

The rest of this paper will be organized as follows. In Section 2, the most recent studies about the cause of accidents and the mechanism of accident factors are reviewed. In Section 3, our research theory and research hypothesis are presented. In Section 4, we present the model of the causal factors chain for MTAs. In Sections 5 and 6, our research is applied to a specific case. The relevant data is collected, analyzed, and applied to the model, and the sensitivity of the model is tested. In Section 7, conclusions are drawn based on our research.

## 2. Literature Review

Increasing industry system safety by reducing infrequent events remains a major challenge to safety scientists. Accident causation methods are broadly applied in the marine traffic field. To study MTAs' occurrence mechanism, the first thing is to understand the causes of the accident and the interaction of the factors that cause the accident [6,7]. Marine accidents result from a combination of complex conditions. Japanese scholars proposed using the marine information structure, holding that the independent action and interaction of human and maritime factors causes most accidents [8]. The complexity of systems and the environments in which humans operate means that the process of safety is not directly forward or linear, but instead is driven by a complex network of relationships and behaviors between humans, technology, and their environment. A new risk management framework is put forward to solve a human control problem and modelling techniques are required to appreciate the direct or indirect operational requirements of systems. The sequence of events reveals a complex interaction between all levels in a socio-technical system spanning strictly physical factors, the unsafe actions of an individual, inadequate oversight, and enforcement [9,10]. In comparison to other accident analysis methods, systems-theoretic accident model and processes (STAMP) uses a functional abstraction approach to model the structure of a system and describe the interrelated functions [11,12]. According to this work-flow, the structure of work systems is hierarchical, in which actors, objects, and tasks are modeled across levels of the complex system and their relationships to each other are linked to explain causal connections. Dynamic work-flows are represented in the framework as inter-dependencies between the vertical integration levels of the system [13]. The functional resonance accident model (FRAM) is different from the traditional model, and used to analyze accidents from the perspective of an internal system operation mechanism or event causal sequence [14]. It does not stick to system structure decomposition and causal factor analysis, and avoids the analysis of accidents into the orderly occurrence of a single associated event, or avoids analysis of the hierarchical stacking of multiple potential factors. Combining safety-I (accident-error oriented) and safety-II (safety-health

oriented) perspectives broadens the understanding of safety management from accident analyses, like causal analysis based system theory (CAST), to hazard analyses, like systems-theoretic process analysis (STPA) [15,16].

Reason (1990) put forward the Swiss cheese model (SCM), the latent and active failures model, and pointed out, for the first time, that an accident is due to the latent defects or vulnerabilities in each part of the system, and that when the defects on each part are lined up, the final cause of the accident can be understood [17]. The model has been criticized for being a reductionist and linear model that fails to account for a holistic representation of systems as dynamic and adaptive, which forms the basis of systems theory [18]. Maintaining the notion of human error as a central concept in an accident causation system disregards the basic fact that the relevant performance is usually carried out by a human-organization factor rather than by an individual. Furthermore, it was shown that about 80% of MTAs are related to human factors [19]. Applications driven by qualitative accident causation models have been improved to investigate human factors in accidents. Subsequently, explorations of the correlation between the causes of MTAs and the consequences of accidents have made significant progress. The main qualitative research investigated the impact of different factors on the outcome of accidents. The relationship among causal factors in accidents has also been studied. Hänninen. (2014) used the directed acyclic graph of the Bayesian network to study the cause of marine accidents [20]. Dai and Wang. (2011) utilized the goal structure notion to analyze the associated rules of human factors to marine accidents [18]. Graziano et al. (2016) used the tracer taxonomy to study human errors in collision accidents [1]. Sotiralis et al. (2016) focused on human centered design aspects to incorporate human factors into ship collisions analysis [21]. Lyu et al. (2018) studied the relationships among safety climate, safety behavior, and safety outcomes in construction workers [22]. The novel drift into failure model (DFM) provides a set of philosophies that explain the nature of drift within a complex system [23]. These embody principles from complexity theory, such as path dependence, non-linearity, and the impact of protective structures [24].

The need to manage human error comes as no great revelation to anyone involved in operations where the consequences of failure are serious. Exploration of the formation methods and mechanism models of human error, and the obtainment of a generalized method for accident investigation, are topics that the industry is constantly studying [25]. Based on the Swiss cheese model, a version of human factors analysis and classification system (HFACS) was established. The HFACS addresses human error at all levels of the system, including the condition of the aircrew and organizational influences [26]. This model is a general human error framework originally developed and firstly tested within the U.S. as a tool for investigating and analyzing the human causes of aviation accidents [27]. It identified several key safety factors that require intervention and proved that the HFACS framework can be a viable tool [28]. Krulak. (2004) proposed a *Maintenance Extension of the HFACS* method (HFACS-ME), and proved that human factors have a significant relationship with mishap frequency and severity in mishaps [29]. Shappell et al. (2007) used the HFACS to put forward a logical method to analyze human factors in the causes of accidents to provide a logical analysis of how accidents occur and how they can be prevented [30]. Celik et al. (2007) sought to integrate those factors into the HFACS system to discover design-based human factors in marine accidents [31].

A general accident model describes the unexpected failures caused by characteristics of a system, where interactions between factors behave in unpredictable ways and produce multiple and unexpected failures. Celik and Cebi (2009) applied the HFACS to qualitatively analyze the human organizational factors (HOF) structure in MTAs [32]. Chen (2013) explored the structural relationship of human factors combined with "why-because" graphs [33]. Hu et al. (2008) used a relative risks model to analyze and evaluate ship navigation safety using the Bayesian belief network [34,35]. Chen et al. (2013) successfully studied the application of the HFACS in coal mines and flight safety, and produced a qualitative list of human factors [36]. Wang et al. (2013) first applied complexity theory to analyze the mechanisms of accident [37]. Within complex systems, the relationships between factors can be described in terms of the interaction between them. Using multiple indicators to reflect latent variables,

and also estimating the relationship between all model factors, is a proposed method to deal with measurement errors, which is more accurate and reasonable than traditional regression methods and is useful to explore the path in an accident causation style. It is necessary to find the principle of path dependence from complexity theory, which has non-linearity, and the impact of protective structures.

Structural equation modeling is a method for testing the relationship between assumed latent variables by using real data collected by researchers [38]. Seo (2005) used the structural equation modeling method to reveal the mechanisms through which the contributory factors of unsafe work behavior influence safety actions of individuals at their workplaces [39,40].

In this paper, we reviewed the research on the mechanisms of MTAs. The HFACS provides a new method for the study of human factors in marine accidents, but a lack of quantitative analysis limits its use. The SEM method makes it possible to quantitatively analyze the relationships among human factors in accidents. Additionally, the lack of a clear path to analyze the causes of MTAs motivated this paper to propose a correlation model of the causal factors chain for MTAs, which is expected to explore the impact of human interactions in the mechanism of accidents.

## 3. Theoretical and Research Hypothesis

### 3.1. HFACS in MTAs

Heinrich classifies the causes of an accident as unsafe behavior of humans, unsafe status of materials, and unsafe conditions of the environment [41]. More and more researchers have begun to study the influence of human factors on accidents. Human factors refer to the harmful effects of human behavior on the normal function or successful performance of the system when completing a specific task.

The HFACS describes human error at each of four levels: The actions of the operators (e.g., bench-level scientists and field investigators in forensics); the preconditions for those actions (i.e., the conditions that influence human behavior); the middle management (i.e., the individuals whose role it is to assign work); and the organization itself [42]. In the maritime field, when using the HFACS for MTAs to analyze human factors in a marine accident [43,44], the basic path of accident formation is described in category I factors, which includes organizational influences, $SL_4$; unsafe supervisions, $SL_3$; preconditions for unsafe acts, $SL_2$; unsafe acts, $SL_1$; and accident, $SL_0$. The accident causal factors and the classification of those factors are defined as shown in Table 1 (category II factors are described as $X_i$, i = 1, 2,... 17). Here, the original framework and structure proposed by Shappell (1997) was reserved, such as $SL_0$ ($X_{17}$), $SL_1$ ($X_{13}$, $X_{14}$, $X_{15}$, $X_{16}$), $SL_2$ ($X_8$, $X_9$, $X_{10}$, $X_{11}$, $X_{12}$), $SL_3$ ($X_4$, $X_5$, $X_6$, $X_7$), and $SL_4$ ($X_1$, $X_2$, $X_3$).

**Table 1.** Relationship of causal factors to marine accidents.

| Symbol | Item of Causal Factors | Symbol | Item of Causal Factors |
|--------|------------------------|--------|------------------------|
| $SL_0$ | Accident | $X_7$ | Violation Monitoring |
| $SL_1$ | Unsafe Acts | $X_8$ | Team factors |
| $SL_2$ | Preconditions for Unsafe Acts | $X_9$ | Individual factors |
| $SL_3$ | Unsafe Supervisions | $X_{10}$ | Material factors |
| $SL_4$ | Organizational Influences | $X_{11}$ | Natural Environment |
| $X_1$ | Resource Management | $X_{12}$ | Physical Environment |
| $X_2$ | Organizational Climate | $X_{13}$ | Slip |
| $X_3$ | Process Safety Control | $X_{14}$ | Lapse |
| $X_4$ | Inadequate Oversight | $X_{15}$ | Mistake |
| $X_5$ | Unsuitable Execution Plan | $X_{16}$ | Violation |
| $X_6$ | Error-Correction Parsing | $X_{17}$ | Accident Consequence |

Based on Swiss cheese model and the theoretical basis for the HFACS, a human structural cheese model can be constructed for a marine traffic accident. As shown in Figure 1, the following hypotheses were made:

**Hypothesis H1:** *SL$_3$ has a significant effect on SL$_4$;*

**Hypothesis H2:** *SL$_2$ has a significant effect on SL$_3$;*

**Hypothesis H3:** *SL$_1$ has a significant effect on SL$_2$.*

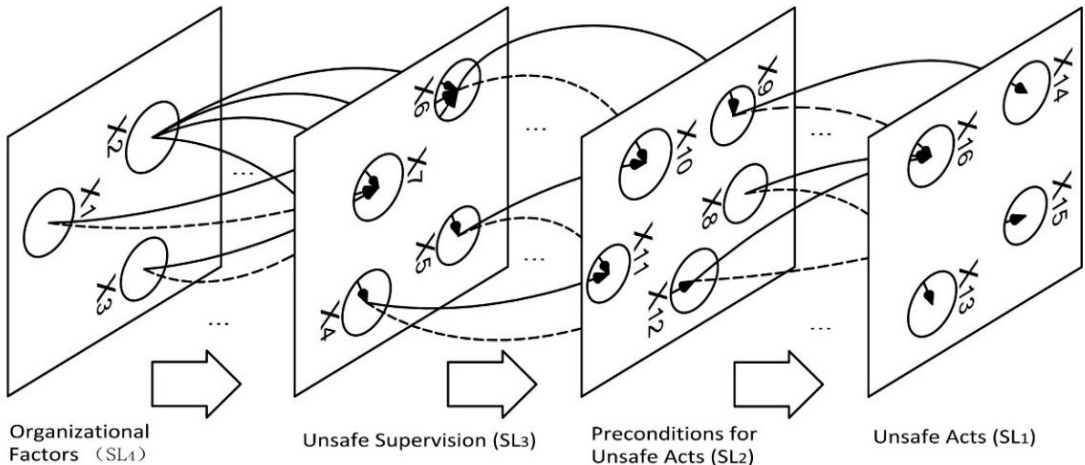

**Figure 1.** Path model of causal factors chain to marine accidents based on the HFACS.

The quantitative relationship among human factors in maritime transportation is discussed thereafter.

*3.2. Causal Factors in MTAs*

Maritime industry stakeholders believe that the human component is a complex, multidimensional proposition that affects maritime safety and marine environmental protection, and includes crew, shore-based management, legislative and law enforcement agencies, shipyards, authorized organizations, and a series of behavioral activities of other relevant parties [35]. All marine accidents are affected and controlled by human factors, ship factors, environmental factors, and management factors. However, the manifestations of system factors vary greatly in different accidents. In order to assist in the implementation of accident case analysis, an accident analysis system needs to be designed to fully define the description and characterization of the cause of the accident. This step relies on historical data and subject-matter experts' analysis from latent sources, such as databases, experiments, simulations, the web, and logical analytical models. Detailed items are shown in Table 2.

**Table 2.** Definitions of causal factors of marine accidents.

| No: | Item | Description and Observation Character |
|---|---|---|
| X$_1$ | Resource management | Ship resource allocation, the allocation of ship resources, including operators, equipment, and facilities, information support, and monitoring, embodied in the suitability of personnel, the seaworthiness of the ship, and the suitability and effectiveness of external supporters. |
| X$_2$ | Organizational climate | The organizational climate can influence employees' events, activities, and procedures, as well as those that may be rewarded, supported, and expected. It can be divided into employees' internal perceptions and team climates. |
| X$_3$ | Process safety control | Process safety refers to how to prevent accidental loss of control and possible traffic accidents caused by installations and facilities during navigation, berthing, and operation process, resulting in damage to employees and ships, environmental damage, and property loss. |
| X$_4$ | Inadequate oversight | No finding in operation arrangements or process issues. Insufficient staff training time, vessel traffic system monitoring failure. |
| X$_5$ | Unsuitable execution plan | Improper arrangement of berths and anchorages, operation plan negligence, operation plan rationality defect. |

| No: | Item | Description and Observation Character |
|---|---|---|
| $X_6$ | Error-correction parsing | Similar accident recurrence |
| $X_7$ | Violation monitoring | Limit cause from draught, weather, ship scale, etc. |
| $X_8$ | Team factors | Crew member's mistake; tug crew error; communication and cooperation negligence. |
| $X_9$ | Individual factors | Illness or bad physiological state; alcoholic beverage; continuous operation, fatigue etc. |
| $X_{10}$ | Material factors | Equipment defects, structural defects, cargo defects, latent defects, overload. |
| $X_{11}$ | Natural environment | Natural disasters, poor visibility, wind currents, tides, surges, navigational environments, waterway bends, navigation aids, navigable waters, fishing areas. |
| $X_{12}$ | Physical environment | Channel curvature; obstacle (including dock or anchorage restrictions). Navigation aid; navigation density; navigable water depth; navigable water width. |
| $X_{13}$ | Slip | Precaution to the natural conditions of the fairway; precaution on ship traffic conditions; precaution to encountering ship behavior; visual hope negligence; navigation instrument not used correctly. |
| $X_{14}$ | Lapse | Navigation operation; avoidance collision behavior; manipulation judgment. |
| $X_{15}$ | Mistake | Emergency treatment; manipulating behavior (an anchorage, by mooring). |
| $X_{16}$ | Violation | Violation operation (relevant ship); violation operation (assisting tugboat); violation operation (pilot); deviation (pilot). |
| $X_{17}$ | Accident consequence | Degree of the consequences of the accident, including near miss. |

### 3.3. Relationship of Causal Factors in MTAs

The relationship of factors is multifactorial. When studying the correlation of human factors in the causes of MTAs, the following aspects are mainly considered:

(1) Positive or negative factors of the correlation coefficient. If the correlation coefficient is positive, there is a positive correlation of the factors; if the correlation coefficient is negative, then there is a negative correlation of the factors.

(2) Scale of the correlation coefficient. For the correlation coefficient, the greater the absolute value, the stronger the correlation of the factors is; if the minimum value is 0, at that time, in general, the factors do not depend on each other.

(3) Rank in the correlation of factors. The interaction of factors is reflected in the relationship of the factors, so some are directly associated, indicating that the factors are direct and influential, but some are indirectly showing secondary effects.

The above associated accident analysis forms the path of the factors. The main content of path analysis is to solve [45]: (a) Path direction; (b) variable relationships of indicators; (c) path load capacity; and (d) whether the model hypothesis is matched.

### 3.4. Path Analysis on Causal Factors of the MTA System

According to complex network theory, the combination of accident factors and their associated relationships is called the accident causal network [46,47]. The node characteristics and associated characteristics in the accident network determine the main performance of the accident network.

The occurrence of complex system safety incidents is not caused by a single risk factor, but is the result of multiple causal risk factors. Corresponding to the accident network, the causal factors of the MTA system are generally not a single node, but an accident chain composed of multiple associated nodes, or an accident network consisting of an accident chain. Therefore, taking into account the dynamic nature of risk, the accident is related to the accident path. However, the accident does not happen overnight, but needs to undergo a series of processes, such as risk emergence, risk transfer, risk coupling, and accident emergence. In this process, there are many risk transfer paths, and the final accident path may be any one of them. One can analyze the risk transmission path of a complex accident system before an accident occurs, and identify the important parameters that affect the risk

transfer. In an accident network, a path with more nodes is a critical path, and a path with fewer nodes is a non-critical path.

The causal path of an accident system includes two parts: The causes of the accident and the relationship of the causes. The path of a maritime causal system can show the beginning and ending of the MTA causal path, namely to express the direct and root causes of the MTA. It shows the causal path of a series of factors interacting with each other before the accident and help better explain the transmission process of the accident cause, thus revealing the evolution mechanism of the accident, and further helping people to take effective measures based on the causal path of vulnerable defects.

## 4. Correlation Model in the Causal Factor Chain for MTAs

Usually, to study the safety of complex systems, it is impossible to test the actual system to observe accident behavior; therefore, one must construct a theoretical model of the complex system. By constructing a corresponding simulation model for the theoretical model, computer simulation can be used to gain an in-depth understanding of a system's performance under different parameters. Traditional multivariate analysis methods, such as complex regression, factor analysis, multivariate analysis of variance, correlation analysis, etc., can only test the relationship between a single independent variable and dependent variable at the same time, and these analytical methods often have deficiency in their theoretical assumptions and application. Factor analysis can reflect the relationship between muti-variables, but it cannot further analyze the causal relationship between variables. While path analysis can analyze the causal relationship between variables, in the actual situation, it is difficult to satisfy the basic assumptions that the measurement error between the variables is zero, the residuals are irrelevant, and the causality is a one-way function. In this paper, a novel method to analyze causal factors is introduced via the network structural equation.

The structural equation model (SEM) is a statistical method that analyzes the relationship among different variables by using a co-variance matrix of variables [45]. The structural equation model integrates path analysis, confirmatory factor analysis, and general statistical test methods to analyze the causal relationship between variables, including the advantages of factor analysis and path analysis. At the same time, it makes up for the shortcomings of factor analysis, taking into account the error factors, and does not need to be limited by the assumptions of path analysis. Based on this, we propose the strong and weak associated path of an accident cause to quantitatively describe the mechanism of the accident.

The purpose of this paper is to find the path to the causes of the accident by finding the relationship among the causes of the accident. This differs from traditional statistical methods because in addition to quantitatively analyzing the effect of a cause on the results, the structural equation model can also quantitatively analyze the relationship between causes, thus this paper uses the structural equation modeling method to decipher the relationships in the causes of an accident.

### 4.1. Methods and Models

The structural equation model includes both the measurement model and the structure model [44,45]. The measurement equation is used to describe the relationship between the observed dependent variable and the latent independent variable. The equation matrices of the measurement model are:

$$x = \lambda_x \xi + \delta \tag{1}$$

$$y = \lambda_y \eta + \varepsilon \tag{2}$$

where among them,

$x$: Vector consisting of observed variables from exogenous latent variables.
$y$: Vector consisting of observed variables from endogenous latent variables.
$\lambda_x$: The strength of association from exogenous observed variables.

$\lambda_y$: The strength of association from endogenous observed variables.
$\zeta$: Unobserved exogenous latent variables.
$\eta$: Unobserved endogenous latent variables.
$\delta$: The error items of the exogenous variables.
$\varepsilon$: The error items of the endogenous variables.

The measurement model is shown in Figure 2.

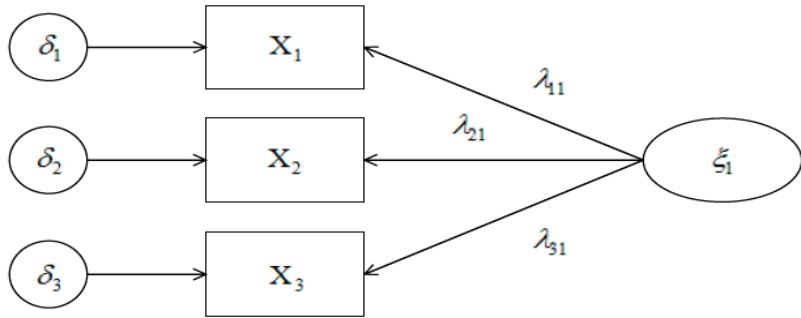

**Figure 2.** The measurement model in structure equation modelling.

Structure equations are used to describe the relationship among latent variables. The equation matrix form of the structure model is:

$$\eta = \beta\eta + \gamma\xi + \zeta \tag{3}$$

where among them,

$\beta$: The relationship between endogenous latent variables.
$\gamma$: The relationship between exogenous latent variables.
$\zeta$: The residual term of the equation, and it represents the portion of the endogenous latent variable that is not interpreted in the SEM.

The structural model is shown in Figure 3.

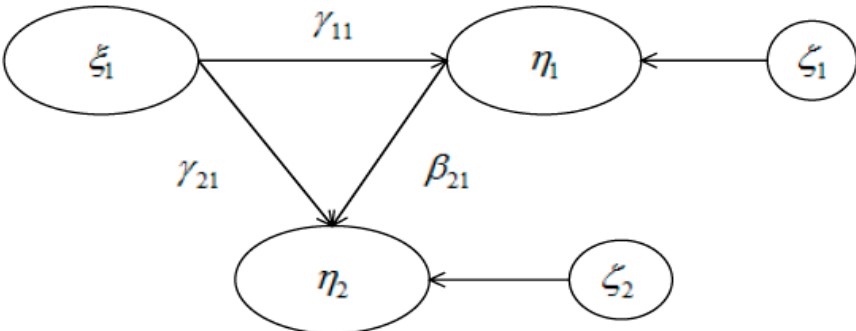

**Figure 3.** The structural model in structure equation modelling.

The above three equations can form a general structural equation model [38]. Each line segment in the SEM has a path coefficient that characterizes the association between two variables connected by the limit. After the path coefficients are normalized, the values range from $-1$ to $+1$. In addition, the values by path factor can be divided into three categories:

(1)     When $0 <$ path coefficient $\leq 1$, it means that there is a positive correlation between variables or one variable has a positive effect on another variable; that is, the function between variables is monotonically increasing.

(2)   When $-1 \leq$ path coefficient $< 0$, it means that there is a negative correlation between variables or one variable has a negative effect on the other variable; that is, the function between variables is monotonously decreasing.

(3)   When the path coefficient is equal to 0, it means that the variables are independent of each other and not related to each other.

### 4.2. Hypothesis Structure Model for the Human Factors of MTAs

The category I factors of the human factors discussed in Section 1 are used as latent variables (indicated by ellipses), and the corresponding category II factors are used as observation variables (indicated by boxes), thus forming a hierarchical classification and hypothesis model of human factors, as shown in Figure 4. $e_i$ is the observation error.

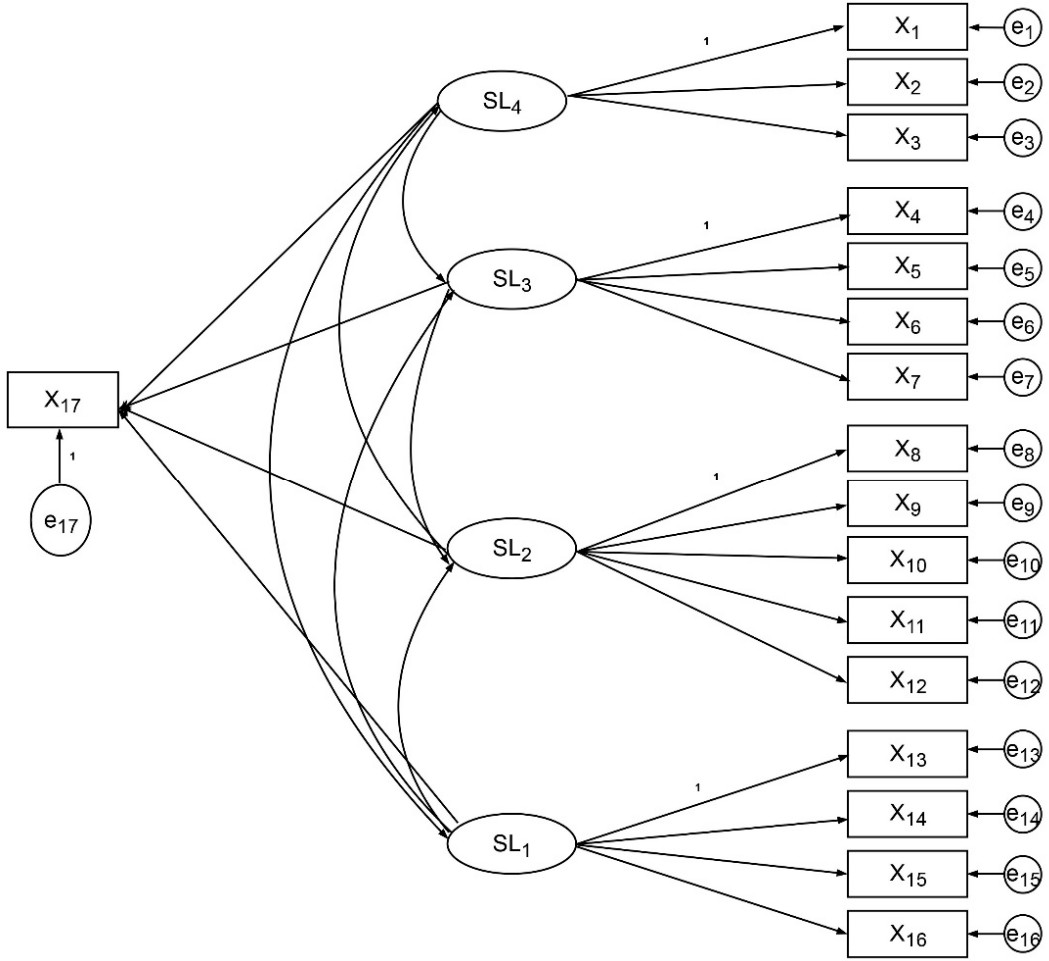

**Figure 4.** Structural Hypothetical Model of Marine traffic accident.

## 5. Case Study

This paper uses the accident case database from 2000 to 2009 in a certain area as an analytic sample [17,34], by the screening and extracting from the database, and combined the SEM hypothesis model with algorithm to apply to the model.

*5.1. Accident Sample Analysis*

5.1.1. Accident Sample Scale

Taking the human error in the area of MTAs as the research object, a total of 894 samples of accidents were introduced. $X_{17}$"accident" as an observation variable is used to examine the effects of different factors on the consequences of the accident. The score of the consequences of the accident depends on the actual level of the collection, including five levels: Incidents, minor accidents, general accidents, major accidents, and serious accidents. They correspond to different accident consequences scores, as shown in Table 3.

**Table 3.** Accident consequence score table.

| Rank | Value | Description |
|---|---|---|
| incidents | 1 | Near miss<br>Hazard<br>An event considered to be worthy of attention |
| minor accidents | 2 | Failure that can be readily compensated by the crew<br>No significant harm to people, property or the environment |
| general accidents | 3 | Local damage to ship<br>Marginal conditions for, or injuries to crew |
| major accidents | 4 | Major casualties excluding total loss<br>Single fatality or multiple severe injuries |
| serious accidents | 5 | Total loss (actual loss and constructive total loss)<br>Many fatalities |

5.1.2. Formatting Causal Factors of the Accident

Among these samples, there were all kinds of consequences, which included 12 incidents, 520 minor accidents, 148 general accidents, 123 major accidents, and 91 serious accidents. The cause analysis of the accidents is the process of determining the cause of the accident and measuring the impact of the accident.

As to the HFACS, human factors are those factors related to people who are involved in the operation of the system. Human factors are beneficial to safety (such as people using their own ingenuity, overcoming the adverse effects of mechanical equipment or harsh environment, etc.), but they can also have a negative effect. As a research object of human factors in MTAs, the negative impact on human safety due to human factors, namely human error, was most important. Detailed information about the observed characters in accident reports was structured and formatted (also shown in Table 2).

Each sample analysis for the causes of the accident is based on the observed characters' items, such as management software, ship (cargo) hardware, environment (including natural conditions, geographical conditions, traffic conditions), and liveware [2]. In the research of human factors in marine traffic safety, the following four interfaces should be analyzed:

(1) Liveware–liveware interface (L–L): The interaction between people in the system, such as leadership, management, communication, and cooperation between people.

(2) Liveware–hardware interface (L–H): The relationship between people and ships, equipment, and other hardware, such as whether the design or layout of the ship or equipment conforms to human characteristics, whether it is convenient for people to manage and maintain the hardware, and to use or operate the hardware.

(3)　Liveware–software interface (L–S): The relationship between people and software, such as whether the information is complete and easy to follow as well as the ease of the operation of the software.

(4)　Liveware–environment interface (L–E): The relationship between humans and the environment, such as whether the working conditions limit human behavior and whether external conditions affect people's judgments.

In the case of the structured accidents' documents, the observed characters in the causes of the accidents were divided into the following seven categories:

(1)　Management items: Maritime administration limit, company management limit.

(2)　Natural items: Natural disasters, poor visibility, wind, tides, surges.

(3)　Channel or terminal items: Navigation loops, channel bends, aids to navigation, navigable waters, chart publications, fishing areas.

(4)　Traffic items: Navigation order, traffic accident, berth anchorage, navigation management.

(5)　Ship cargo items: Structural defects, equipment defects, cargo defects, latent defects, large workload.

(6)　Personnel involved items: The tugboat operator, the ship operator, and the outboard operator.

(7)　Crew items: Violation operation, negligence of route planning, negligence of navigation operation, negligence in avoidance of collision, negligent manipulation, emergency-handling, communication and cooperation negligence.

According to the different effects of the observed character on the outcome of these accidents, the factors' influence levels are divided into four grades:

Level I, the factor may not impact the accident outcome, no effect.
Level II, the factor may partly impact the accident outcome, involved.
Level III, the factor may mainly impact the accident outcome, mainly.
Level IV, the factor may apparently impact the accident outcome, directly.

*5.2. Data Acquisition and Reliability Analysis*

In order to enable the fitting of the collected data into the hypothesis model, the collected accident factors were quantified according to the level of the impact on the consequences of the accidents. In this paper, to evaluate and synthesize the samples collected, a workshop was conducted with subject-matter experts in accident analysis and systems thinking. Furthermore, the data in accident causation were measured by the "Likert scale", using a five-level scale.

First, quantitative data assignment was used for the extent of each factor's effect. According to the level of impact, the rating is separately defined, such as no effect, 5; involved, 4; mainly, 2; directly, 1. Regarding how the accident is described, for example, those that are described as a general accident, the detailed influence factors, which result in a certain accident, include observed characters, such as "Non finding in operation arrangements or process issues", "Insufficient staff training time", and "VTS monitoring failure" (variable in Table 2). These factors affect the accident at different levels of influence as discussed above, namely, "directly"," involved", and "no effect", respectively. That means the score is 1, 4, and 5, respectively. Each accident sample can be described by the influence factors.

Second, the score of the $X_i$ ($i$ = 1, 2,... 16) accident causal factors depends on the minimum score among the corresponding observed characters collected. As to the case statement above, those three observed characters involving "Inadequate oversight" were numerically analyzed, and the lowest score is measured as 1, which means "directly". Therefore, $x_4$ "Inadequate oversight" is measured as 1. All the structured observed characters in the accident reports were formatted to numerical analysis data. The tested data statistics are shown in Table 4.

**Table 4.** Tested data from the accident database.

| Case No: | $X_1$ | $X_2$ | $X_3$ | $X_4$ | $X_5$ | $X_6$ | $X_7$ | $X_8$ | $X_9$ | $X_{10}$ | $X_{11}$ | $X_{12}$ | $X_{13}$ | $X_{14}$ | $X_{15}$ | $X_{16}$ | $X_{17}$ |
|---|---|---|---|---|---|---|---|---|---|---|---|---|---|---|---|---|---|
| 1 | 5 | 4 | 4 | 5 | 1 | 5 | 1 | 1 | 1 | 4 | 2 | 4 | 1 | 4 | 4 | 1 | 3 |
| 2 | 4 | 4 | 5 | 5 | 5 | 5 | 5 | 5 | 1 | 5 | 2 | 4 | 1 | 4 | 4 | 5 | 5 |
| 3 | 4 | 5 | 5 | 5 | 4 | 4 | 4 | 1 | 1 | 5 | 4 | 5 | 1 | 4 | 4 | 5 | 4 |
| 4 | 4 | 4 | 4 | 5 | 5 | 4 | 5 | 5 | 5 | 5 | 5 | 4 | 5 | 4 | 5 | 5 | 4 |
| 5 | 5 | 5 | 4 | 4 | 5 | 5 | 4 | 1 | 1 | 5 | 5 | 4 | 1 | 4 | 5 | 4 | 4 |
| 6 | 4 | 4 | 5 | 4 | 5 | 4 | 4 | 1 | 1 | 5 | 2 | 4 | 1 | 4 | 5 | 4 | 4 |
| 7 | 4 | 5 | 4 | 5 | 5 | 4 | 5 | 4 | 5 | 5 | 5 | 5 | 5 | 5 | 4 | 4 | 4 |
| 8 | 5 | 4 | 5 | 4 | 5 | 4 | 5 | 5 | 1 | 4 | 2 | 5 | 1 | 4 | 4 | 4 | 5 |
| 9 | 5 | 5 | 5 | 5 | 1 | 5 | 1 | 1 | 1 | 4 | 5 | 5 | 1 | 4 | 5 | 1 | 3 |
| 10 | 5 | 5 | 5 | 4 | 1 | 4 | 1 | 1 | 1 | 5 | 4 | 5 | 1 | 5 | 4 | 1 | 3 |
| ———— | | | | | | | | | | | | | | | | | |
| 891 | 4 | 5 | 4 | 4 | 5 | 5 | 5 | 1 | 1 | 2 | 4 | 4 | 1 | 4 | 4 | 4 | 2 |
| 892 | 4 | 5 | 4 | 5 | 5 | 5 | 5 | 1 | 1 | 2 | 5 | 5 | 1 | 4 | 5 | 4 | 2 |
| 893 | 4 | 4 | 4 | 4 | 1 | 5 | 1 | 1 | 1 | 5 | 2 | 4 | 1 | 5 | 5 | 1 | 1 |
| 894 | 4 | 5 | 5 | 5 | 1 | 4 | 1 | 1 | 1 | 5 | 4 | 4 | 1 | 4 | 5 | 1 | 1 |
| μ | 4.1 | 4.2 | 4.1 | 4.2 | 4.0 | 4.1 | 2.7 | 2.9 | 2.3 | 4.1 | 3.7 | 4.4 | 2.1 | 4.5 | 4.5 | 4.0 | 4.3 |
| σ | 0.8 | 0.7 | 0.7 | 0.7 | 1.3 | 0.8 | 1.3 | 1.7 | 1.7 | 0.9 | 1.1 | 0.4 | 1.6 | 0.4 | 0.4 | 1.3 | 1.1 |

The collected accident factors were categorized according to the literature [17,34], and finally the data was integrated into the 16 major accident factors. Thereby, the scoring of the 16 accident factors (variable in Figure 5) depends on the corresponding minimum score among the accident factors collected.

In addition to the correlation of factors in different MTAs, the impact of different factors on the consequences of the MTA was also analyzed. Therefore, the observation variable of "Accident consequence" ($X_{17}$) was added to examine the influence of different factors on the consequences of accidents.

## 5.3. Model Fitting and Correction

The paths that did not conform to the SEM hypothesis are as follows: (a) The path of the error term of the observed variable to the latent variable; (b) the path of the observed variable to other observed variables; (c) the error term of the observed variable for other observations; (d) the path of influence of the variable; (e) the path of the error term of the observed variable to the error term of other observed variables.

When the model is changed, the researcher should add new paths one by one instead of adding multiple paths all at once. The processed data were fitted with the hypothetical model, and the model was modified with the output of the modification indices. The resulting path dependency is shown in Figure 5.

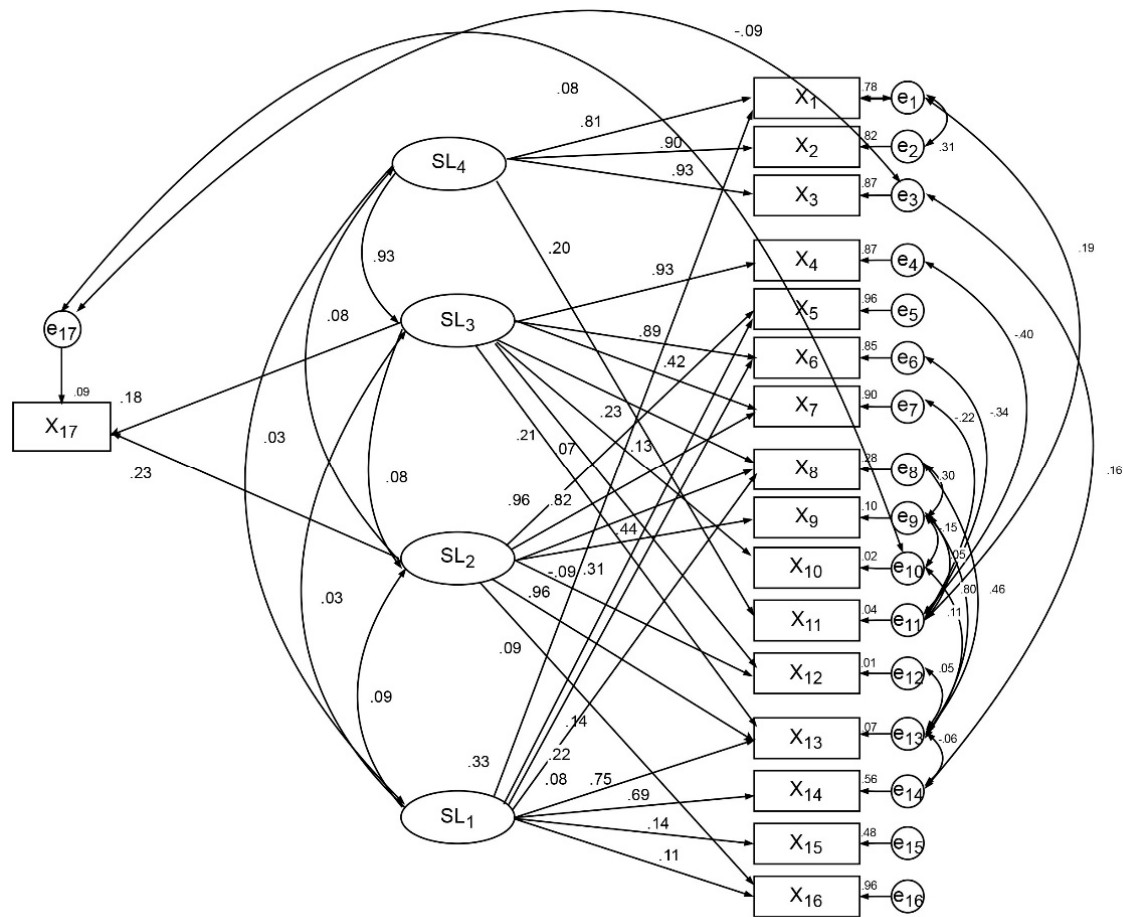

**Figure 5.** Path diagram of marine accidents formation using structural equation model simulation.

*5.4. Reliability Analysis in Path Dependency*

An analysis of the reliability of the sample data table should be performed before fitting the sample data to the hypothetical model [38]. Cronbach's alpha coefficient (CA) is a measure of the intrinsic consistency of a set of data used to determine whether the set of data represents the same attitude tendencies and whether it can form an attitude measurement index.

Cranach's alpha test was performed on the observation variables to measure a set of hypothetical "internal consistency" coefficients (Byrne, 2009) to judge whether this group of hypotheses represented the same tendency of attitude and whether it constituted an attitude measurement index.

In general, if the CA is greater than 0.7, this indicates that the data had good reliability. When the CA is below 0.7, the entries in the data may represent different dimensions and need to be filtered.

The results show that after deleting some of the items, the check coefficient values of the observed variables are all above 0.7, and the overall reliability value reaches 0.797, indicating that this figure has good reliability.

Data statistics are shown in Table 4, which shows the mean and standard variation of each variable.

Since the modified model used in this paper has some differences with the theory, it is necessary to test the sensitivity of the model in order to verify whether the modified model used in this paper is applicable to different types and sizes.

The critical ratio (C.R.) is used to test the significance of the evaluation of each parameter in the model [45]. The critical ratio is the proportion of the evaluation of the parameter estimate to its standard deviation. When the significance level is 0.05, it means that the parameter evaluation is not significantly equal to 0, and the null hypothesis can be rejected if the absolute value of C.R. is greater than 1.96. The calculation results are presented in Table 5.

**Table 5.** Data of the critical ratio of variables.

| Hypothesis | Estimate Value | Critical Ratio | Conclusion |
|---|---|---|---|
| H1: $SL_3 -> SL_4$ | 0.944 | 33.727 | Significant influences exist, defined hypothesis is true |
| H2: $SL_2 -> SL_3$ | 0.077 | 2.175 | Significant influences exist, defined hypothesis is true |
| H3: $SL_1 -> SL_2$ | 0.125 | 2.921 | Significant influences exist, defined hypothesis is true |

The goodness-of-fit index of the amended model is shown in Table 6.

From Tables 5 and 6, it is evident that the goodness-of-fit index of the model meets the criteria, indicating that the model and the data fit well.

**Table 6.** Data of the variables via SEM simulation.

| Evaluation Index | Estimate Value | Adaptation Standard |
|---|---|---|
| *Absolute index* | | |
| $x^2$ Significant probability value | 0.281 | >0.05 |
| Goodness-of-fit index (GFI) | 0.989 | >0.90 |
| Adjusted goodness-of-fit index (AGFI) | 0.980 | >0.90 |
| Root mean square residual (RMR) | 0.031 | <0.05 |
| Root mean square error of approximation (RMSEA) | 0.010 | <0.05 |
| *Relative index* | | |
| Normal fit index (NFI) | 0.993 | >0.90 |
| Relative fitness index (RFI) | 0.988 | >0.90 |
| Incremental fit index (IFI) | 0.999 | >0.90 |
| Tracker—Lewis index (TLI) | 0.999 | >0.90 |
| Comparative-fit index (CFI) | 0.999 | >0.90 |
| *Parsimony index* | | |
| Parsimony goodness-of-fit Index (PGFI) | 0.552 | >0.50 |
| Parsimony-adjusted (PNFI) | 0.629 | >0.50 |
| $x^2/nf$ (NC) indicating the degree of minimalist fit | 1.088 | $1 < NC < 3$ |

It can be seen from Table 5 that the path coefficient of $SL_4 -> SL_3$ is 0.94 and the *t*-check value is 33.727; the path coefficient of $SL_3 -> SL_2$ is 0.08 and the *t*-check value is 2.175; and the path coefficient of $SL_2 -> SL_1$ is 0.13 and the *t*-check value is 2.921. These indicate that the *H1*, *H2*, and *H3* hypotheses are true and have a significant positive relationship. This proves the correctness of the HFACS-MTA framework from a quantitative point of view.

*5.5. Sensitivity Analysis of the HFACS-MTA Based on the SEM Model*

Sensitivity analysis was used to qualitatively or quantitatively analyze changes in the model results when model parameters or samples change. It classified the collected documented cases according to different types of accidents (such as collisions, grounding, fires, etc.), which fitted different types of accident data to the revised model of Figure 5, and a model analysis of the changes in the goodness-of-fit index and the estimated parameters was carried out, in order to test the reliability and stability of the model. The post-test data prove that: Although the significance level of the chi-square value obtained by fitting the modified model with the test sample did not reach the goodness-of-fit

index, other fitness indexes met the requirements, and most of the path coefficients shown by the model were consistent. Therefore, the modification model of the MTA causal path is stable and suitable for applications to samples under different conditions, and can provide guidance in those situations.

There were some differences between the model results and the HFAC-MTA in the corresponding relationship of the category I factors and category II factors, as presented in Table 7.

**Table 7.** Factors' correlation characters via SEM simulation.

| Correlation Mode | | | Standardized Path Coefficient |
|---|---|---|---|
| $SL_4$ | -> | $X_{11}$ | 0.24 |
| $SL_3$ | -> | $X_5$ | No significant effect |
| | | $X_{13}$ | 0.21 |
| $SL_2$ | -> | $X_5$ | 0.94 |
| | | $X_7$ | 0.82 |
| $SL_1$ | -> | $X_1$ | 0.27 |
| | | $X_5$ | 0.16 |
| | | $X_6$ | 0.23 |
| | | $X_8$ | 0.08 |
| | | $X_{10}$ | 0.09 |

Table 7 shows that:

(1) Organizational influences, $SL_4$, are not only related to the three types of human factors in the theory, but also related to the natural environment.

(2) There is no significant correlation between unsafe supervision, $SL_3$, and unsuitable execution plan, $X_5$, in HFACS theory, but there is a correlation with slip, $X_{13}$.

(3) The preconditions for unsafe acts, $SL_2$, are related to unsuitable execution plan, $X_5$, and violation monitoring, $X_7$.

(4) There are correlations between unsafe acts, $SL_1$, and resource management, $X_1$, unsuitable execution plan, $X_5$, error-correction parsing, $X_6$, team factors, $X_8$, and material factors, $X_{10}$.

## 6. Path Analysis and Discussion

Path analysis is used to test the hypothesised relationship of observation variables or indicator variables. The purpose of path analysis is to check the accuracy and reliability of the hypothetical model and analyze the relation intensity of different variables. Figure 5 mainly shows the path diagrams of latent variables and latent variables with their corresponding observed variables. However, the relationship among observed variables could not be obtained, and there is a correlation in the measurement error items of the model. The correlation between the two measurement error items indicates that there is a certain degree of latent correlation between the corresponding two measurement variables. From this, the MTA causal system path diagram is as shown in Figure 6 (only select the part in which the normalized path coefficients are greater than 0.2 between category I factors and category II factors).

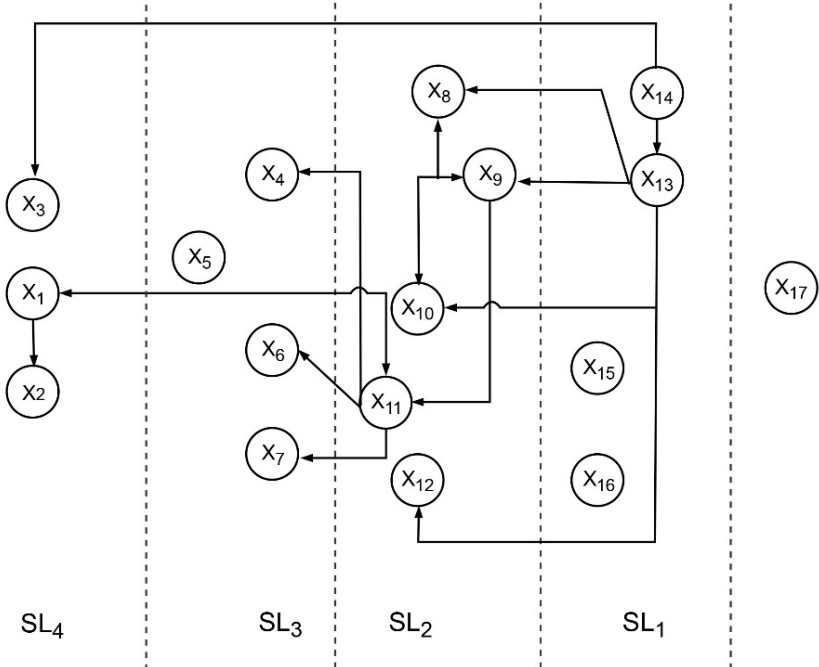

**Figure 6.** Path and trace representation of the causal in marine traffic accident network.

Figure 6 presents some path dependencies that may lead to accidents, such as:

Path dependency I (PD-I): Resource management—natural environment—individual factors—slip.

Path dependency II (PD-II): Organizational climate—resource management—natural environment—error-correction parsing.

Decision-makers can find the influence and mode of action in the causes of MTAs based on these path dependencies. For example, the PD-I link indicates that there is an interaction between the "resource management" and "natural environment", "natural environment" and "individual factors", and "individual factors" and "slip" and these interactions eventually result in accidents.

- The "natural environment" is the important reason for the entire accident system, and it is the key link between the previous factor and the next.
- "Resource management" has a prominent position in the organizational influence level (root cause) and is highly relevant.
- "Process safety control" directly affects the "slip" of unsafe human acts.

Therefore, the decision-maker can strengthen the control and management of four structural factors of the causal path to avoid interactions and ultimately prevent an accident from occurring. It is also possible to intervene in only some of the key items, to cut off the progression of the causal path and eventually avoid an accident.

(1) The organizational influences, $SL_4$, corresponding to category II human factors are resource management, organizational climate, process safety control, and natural environment. Category II human factors corresponding to unsafe supervisions, are: Error-correction parsing, inadequate oversight, violation monitoring, team factors, and slip.

(2) The preconditions for unsafe acts, $SL_2$, corresponding to category II human factors are violation monitoring, team factors, unsuitable execution plan, individual factors, and violation.

(3) The unsafe acts, $SL_1$, corresponding to category II human factors are resource management, error-correction parsing, lapse, and mistake. Among them, resource management, error-correction parsing, team factors, and violation monitoring distribution are related to two category I human factors.

(4)     After comparing the four levels of the HFACS framework, organizational influences, $SL_4$, preconditions for unsafe acts, $SL_2$, and unsafe acts, $SL_1$, were detected to strongly contribute to marine accident risks. This implies that organizational and individual factors should be emphasized instead of unsafe supervision, $SL_3$, considerations. This study further identified that the factors at the preconditions for unsafe acts level are most influential to marine accident risks among all factors at the HFACS levels, and the unsafe supervisions level influences marine accident consequences.

(5)     From the path of the accident, there are simple chains, complex chains, and system networks [46,47]. The accident path is a simple chain described by the domino model, Swiss cheese model, and the HFACS. The domino model considers that the accident causes the dominoes represented by each module to fall down one after another so that an accident will occur. This logic model was clear, but such a simple linear description cannot truly reflect the nonlinear interactions of various factors present in complex social technology systems. The path of the accident described by the trajectory crossover model is a complex chain, in which, such as in this model, two parallel paths are proposed to lead to the accident. This study has involved the thinking mode of system theory on the HFACS to describe the path of the system network about an accident. It is thought that there are both hierarchical and causal relationships between the causes of accidents, and the interactions are mixed to form a network, which is closer to the real material world.

## 7. Conclusions

The formation of MTAs is complex, but the degree of influence and the mode of action of factors in the cause system are different. The strength of the correlation of the factors determines the path of the accident. Example verification shows that there are different correlations of various factors in the HFACS, and the observed variables manifested form conforms to the path dependency mode. Resource management factors in the sub-hierarchy of organizational influences have a prominent position in the accident formation and a strong correlation.

(1)     The HFACS-MTA generic texture hypothesis paradigm based on the SEM can develop system pathway maps between the latent (independent) variable and observed (dependent) variable, which could allow quantitative study of interrelationships in various causes. The hypothesis model application shows that the relationship of human factors in the MTA is consistent with the HFACS, and the direction of human error in the MTA is in the order of organizational influences, unsafe supervisions, and preconditions for unsafe acts, and finally passed on to unsafe acts. The mutual influences in the factors of the accident causes are obviously different.

(2)     Structural equation modeling is a powerful research tool in the field of safety sciences, but the establishment of related models relies on the knowledge of relevant scientific fields. The setting of the implicit variables of the structural equations of accident causation theory and the setting of the relationship between hidden variables are the theoretical knowledge base of the maritime field. The setting and measurement of the measured variables corresponding to the hidden variables also have their theoretical basis. The structural equation model is only a mathematical expression of the theoretical knowledge base of the relevant scientific field, and it provides a tool for us to study related safety sciences.

(3)     We have seen that in recent times, the theory of safety-oriented causation based on system theory has greatly changed and developed an understanding of the formation of traditional accidents. In particular, the characteristics of safety are seen as an emergence of systems, with safety issues as a matter of control. The cause of the accident is not only to describe the components in the system structure, but also to explain the interaction and coupling between the causal factors. This paper believes that a certain mathematical algorithm is used to analyze the degree of cross-linking between factors, describe the process of controlling between factors, and then determine the path

of accident formation. This is a quantitative demonstration of the cheese model, revealing the path dependence of management defects in the field of marine safety affecting human behavior.

(4) We also see that to study the safety problems of complex marine traffic systems, a theoretical model of a complex system needs to be constructed, and thus an accident cause structural hypothesis was proposed. Appropriate algorithms for the theoretical human-machine-control model were used to understand the safety performance of marine traffic systems under different parameters through mathematical analysis. Accident databases providing manifold data were only measured, but were subjective especially in relation to the assessment of human failures and the question of how consistent the database is, which remains a critical issue. Combined with big data ideas and intelligent prediction theory, it provides an important basis for risk pre-warning and accident prevention. This will be a problem that will require further research.

**Author Contributions:** Conceptualization, S.H. and X.G.; methodology, S.H. and Y.X.; software, X.G.; validation, S.H. and X.Z.; writing—original draft preparation, Z.L.; writing—review and editing, S.H.

**Funding:** This work was supported by the Shanghai International Port (Group) Co., Ltd. Technology Innovation Project (2017) (Pilot Station_17KY-04B-31Z).

**Acknowledgments:** We appreciate the data support from Fujian Maritime Safety Administrator (MSA), China. We would also like to acknowledge the insight contributions from two anonymous reviewers whose thoughtful comments have helped to improve an earlier version of this paper.

**Conflicts of Interest:** The authors declare no conflict of interest.

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
