# Peer review of "Path Analysis of Causal Factors Influencing Marine Traffic Accident via Structural Equation Numerical Modeling"

_jmse, doi:10.3390/jmse7040096_

Reviewer 1 Report

The paper presents a method for quantitative presentation of maritime traffic accidents and their causes. The topic is important as a ship accident can have major consequences. The paper is quite easy to read and quite extensive. However, there are some issues that should be addressed before accepting the paper to be published.

Safety science is quite recent field and because of this, system safety and accident causation research still have several theoretical schools of thought. The safety approach adapted in the paper, HFACS and its underlying CREAM methodology, belong to so-called "Safety I" class (2nd generation models). It has been argued that the more recent "Safety II" approach (3rd gen models or the systemic approach) is currently the dominant one (see e.g. "Back to the future: What do accident causation models tell us about accident prediction?" by Grant et al.) and "Safety I" is considered somewhat old-fashioned. Thus, the paper should acknowledge and describe both of these theoretical approaches to safety and then justify more clearly why the selected method and "Safety I" type of approach was chosen. Furthermore, some sentences in the Introduction chapter and chapter 3.4 give the impression that ideas following the "Safety I" views to accident causation are widely accepted facts, when in fact "Safety II" scholars disagree on these (for example, accident chain as a concept). Thus the paper should provide references to work which the views adapted in the paper are based on.

When checking some of the references used in the literature review, I was not unable to find support for the propositions, e.g. the paper by Goerlandt & Montewka (2015) does not discuss accident occurrence mechanisms and causal factor interactions at all, and the paper by Dekker (2002) does not state that HFACS is one of the most useful methods (or at least I can find such statement in Dekker's paper). Has there been some errors in referring to correct papers or what might be the reason for these?

In chapter 3, it is not clear to me, how the causal factors presented in Table 2 have been selected (do they come e.g. from Chauvin et al, Wu et al, some IMO documentation or are they defined by the authors) - could you please clarify this?

In chapter 4, the motivation given to apply SEM could also hold for applying causal Bayesian networks, which are very similat to SEM and have been used in maritime traffic accident analysis quite much (see e.g. Hänninen 2014 - the paper which the authors already use as a reference). So please clarify the advantages of using SEM a bit more. Also, please clarify what you mean by "the traditional method" to which you compare SEM.

In chapter 5, please describe the used data set a bit, as I was not able to find its description from the paper used as a reference either (Hu et al. 2012).

Chapter 5.1 and Table 1: how has these levels been decided (e.g. that a case with a single fatality belongs to major accident class)? Please clarify. Also, why are the accident cause categories presented in Chapter 5.1 lines 294-305 different from Table 2 categories and how have these seven categories been defined? Later in Chapter 5.1 the 16 categories from Table 2 are again used instead of the seven ones, this is a bit confusing.

Chapter 5.2: it is very difficult for me to follow how the scoring works. This part should be explained better. Also, more information is needed on how the cases were analysed and scored: 894 cases are a lot of work to be analyzed by a human, how was it done? Also, it has been shown that accident analysis is prone to biases (e.g. "What-You-Look-For-Is-What-You-Find – The consequences of underlying accident models in eight accident investigation manuals" by Lundberg et al) - who did the analysis?

Chapter 5.4 is quite confusing. For example, based on the text it remains unclear to me why Cronbach's alpha test is done. Could you please revise and clarify this part? Also, the link between parameter sensitivity analysis and critical ratio is not clear, and the model fit part does not mention whether it was done to the training data or test data etc.

Chapters 6 and 7. More discussion on the findings from the model and what can be said regarding maritime traffic accidents and safety based on those would be needed. Now the focus is a bit too much on the model approach details.

Author Response

first of all, We would also like to acknowledge the insight 558 contributions from the anonymous reviewers whose thoughtful comments have helped to improve an earlier version of this paper.

About the chapter II, we updated more detail.

 In particular, we have added five articles on the status quo and progress of theoretical research on the cause of accidents. Emphasis is placed on the development of the understanding of the cause of accidents after the introduction of the complexity system. Safety I and Safety II principals were analysized. based on safety-I, we put forward to study the path dependence from complexity theory, which has the non-linearity and the impact of protective structures. 

Chapter 3, about table 2, we add the description: 

Here the original framework and structure proposed by Shappell (1997) was reserved, such as SL1 (X13, X14, X15, X16), SL2 (X8, X9, X10, X11, X12),SL3 (X4, X5, X6, X7), and SL4 (X1, X2, X3).

and

All marine accidents will be affected and controlled by human factors, ship factors, environmental factors and management factors. However, the manifestations of system factors vary greatly in different accidents. In order to assist in the implementation of accident case analysis, an accident analysis system needs to be designed to fully define the description and characterization of the cause of the accident. This step relies on historical data and subject-matter experts analysis from the latent sources, such as databases, experiments, simulations, webs and logical analytical models.

Chapter 4, about SEM, we add the description: 

The Structural Equation Model(SEM) is a statistical method that analyzes the relationship among different variables by using a covariance matrix of variables. The structural equation model integrates path analysis, confirmatory factor analysis and general statistical test methods to analyze the causal relationship between variables, including the advantages of factor analysis and path analysis. At the same time, it makes up for the shortcomings of factor analysis, taking into account the error factors, and does not need to be limited by the assumptions of path analysis. Based on this, we propose the strong and weak associated path of accident cause to quantitatively describe the mechanism of the accident.

In chapter 5, about the data,we add the description: 5.1.2

As to HFACS, human factors are those factors related to people in the operation of the system. Human factors are beneficial to the safety side (such as people exert their own ingenuity, overcome the adverse effects of mechanical equipment or harsh environment, etc.), but also can have a negative effect. As a research object of human factors in MTA, the negative impact on human safety due to human factors, namely Human Error was more important. The detailed information about the observed characters in accident reports were structured and formatted(also shown in Table 2).

Each sample analysis for the causes of the accident is based on observed characters items, such as management software , ship (cargo) hardware, environment(including natural conditions, geographical conditions, traffic conditions), and liveware (Xi et al., 2017). 

In the case of structured accidents’ documents, the observed characters in causes of the accidents are divided into the following 7 categories.

In chapter 5, about the data analysis,we add the description: 

First, quantitative data assignment is used for the extent of each factor's effect. According to the Level of impact, the rating is separately defined. Such as no effect, 5; involved, 4; mainly, 2; directly, 1. As to how the accident is described , for example, those which are described as a general accident, the detail influence factors which result to a certain accident includes observed characters such as “non finding in operation arrangements or process issues”, “Insufficient staff training time” and “VTS monitoring failure” (variable in Table 2). These factors effect the accident at different levels of influence as discussed above, namely, “directly”,” involved” and “no effect” respectively. That means the score is 1, 4, 5 respectively.Each accident sample can be described by the influence factors. 

Second,accident causal factors depends on the minimum score among the corresponding observed characters collected. As to the case statemented above, those 3 observed characters involved “Inadequate Supervision” were numerical analyse, and the lowest score is measured as 1, which means “directly”. Therefore, x4“Inadequate Supervision” is measured as 1. All the structured observed characters in accident reports were formatted to numerical analysis data. The tested data statistics are shown in Table 4.

In chapter 5, about the case study,we amend the description: 

5.4 Reliability analysis in path dependency

5.5 Sensitivity analysis of HFACS-MTA based on SEM model 

Chapters  7, we amend the description: 

Structural equation modeling is a powerful research tool in the field of safety sciences, but the establishment of related models relies on the knowledge of relevant scientific fields. The setting of the implicit variables of the structural equations of accident causation theory and the setting of the relationship between hidden variables have the theoretical knowledge base of the maritime field. The setting and measurement of the measured variables corresponding to the hidden variables also have their theoretical basis. The structural equation model is only a mathematical expression of the theoretical knowledge base of the relevant scientific field, and it provides a tool for us to study related safety sciences.

Reviewer 2 Report

1. The purpose of the study should be clearly stated in the abstract.

2. Line 40 ~ 41 : References should be listed.

3. Line 49 ~ 50 : References should be listed.

4. Line 90: Please check whether "graphs, et al" is typo.

5. Line 66 (Literature Review): A review of prior studies is to find limitations that were not covered in previous studies and to show that this study has overcome its limitations and has achieved new research achievements. There’s a lot of prior studies, but have little mention of differentiation. Please make it clear.

6. Line 116 : References should be listed.

7. Line 134 : Please indicate Category I and II in Table 1.

8. Line 134 : Please provide the reason for selecting items from X1 to X17.

9. Line 136 : Please provide the reason of the classification of SL4 (X1, X2, X3) .... SL1 (X13, X14, X15, X16).

10. Line 141 : References should be listed.

11. Line 328 : Is the scale of X17 in Table 4 the standard of Table 3? Please note this. If it is the criterion in Table 3, X17 is serious as the level goes up, and X1 ~ X16 is not influenced from 1 to 5. I am curious as to whether the analysis is possible

12. Please describe in detail how the identification of the Likert scale of Table 4 has progressed. Please describe the criteria for each factor for the Likert scale.

And I think it would be quick to express an example of an accident case.

13. 346 Line: Please describe the analysis for Figure 5 in detail

14. Line 355 : References should be listed.

15. Line 364 : (Figure 6) It is difficult to accurately determine mean and standard deviation with cumulative graph. Please reprint the graph.

16. Line 446 : (Conclusion) The entire contents are too short for description. Please describe the conclusion of the research in a little more detail.

Author Response

first of all, We would also like to acknowledge the insight 558 contributions from the anonymous reviewers whose thoughtful comments have helped to improve an earlier version of this paper.

About the chapter II, we updated more detail. In particular, we have added five articles on the status quo and progress of theoretical research on the cause of accidents. Emphasis is placed on the development of the understanding of the cause of accidents after the introduction of the complexity system. Safety I and Safety II principals were analysized. based on safety-I, we put forward to study the path dependence from complexity theory, which has the non-linearity and the impact of protective structures. 

Chapter 3, about table 2, we add the description: 

Here the original framework and structure proposed by Shappell (1997) was reserved, such as SL1 (X13, X14, X15, X16), SL2 (X8, X9, X10, X11, X12),SL3 (X4, X5, X6, X7), and SL4 (X1, X2, X3).

and

All marine accidents will be affected and controlled by human factors, ship factors, environmental factors and management factors. However, the manifestations of system factors vary greatly in different accidents. In order to assist in the implementation of accident case analysis, an accident analysis system needs to be designed to fully define the description and characterization of the cause of the accident. This step relies on historical data and subject-matter experts analysis from the latent sources, such as databases, experiments, simulations, webs and logical analytical models.

In chapter 5, about the data,we add the description: 5.1.2

As to HFACS, human factors are those factors related to people in the operation of the system. Human factors are beneficial to the safety side (such as people exert their own ingenuity, overcome the adverse effects of mechanical equipment or harsh environment, etc.), but also can have a negative effect. As a research object of human factors in MTA, the negative impact on human safety due to human factors, namely Human Error was more important. The detailed information about the observed characters in accident reports were structured and formatted(also shown in Table 2).

Each sample analysis for the causes of the accident is based on observed characters items, such as management software , ship (cargo) hardware, environment(including natural conditions, geographical conditions, traffic conditions), and liveware (Xi et al., 2017). 

In the case of structured accidents’ documents, the observed characters in causes of the accidents are divided into the following 7 categories.

In chapter 5, about the data analysis,we add the description: 

First, quantitative data assignment is used for the extent of each factor's effect. According to the Level of impact, the rating is separately defined. Such as no effect, 5; involved, 4; mainly, 2; directly, 1. As to how the accident is described , for example, those which are described as a general accident, the detail influence factors which result to a certain accident includes observed characters such as “non finding in operation arrangements or process issues”, “Insufficient staff training time” and “VTS monitoring failure” (variable in Table 2). These factors effect the accident at different levels of influence as discussed above, namely, “directly”,” involved” and “no effect” respectively. That means the score is 1, 4, 5 respectively.Each accident sample can be described by the influence factors. 

Second,accident causal factors depends on the minimum score among the corresponding observed characters collected. As to the case statemented above, those 3 observed characters involved “Inadequate Supervision” were numerical analyse, and the lowest score is measured as 1, which means “directly”. Therefore, x4“Inadequate Supervision” is measured as 1. All the structured observed characters in accident reports were formatted to numerical analysis data. The tested data statistics are shown in Table 4.

Chapters  7, we amend the description: 

Structural equation modeling is a powerful research tool in the field of safety sciences, but the establishment of related models relies on the knowledge of relevant scientific fields. The setting of the implicit variables of the structural equations of accident causation theory and the setting of the relationship between hidden variables have the theoretical knowledge base of the maritime field. The setting and measurement of the measured variables corresponding to the hidden variables also have their theoretical basis. The structural equation model is only a mathematical expression of the theoretical knowledge base of the relevant scientific field, and it provides a tool for us to study related safety sciences.

Reviewer 3 Report

Dear authors,

first of all, congratulations to a really interesting and valueable paper. The aim of your work is quite ambitious. Applying latest research and state of the art developments in accident analysis and prevention os of highest importance and relevance. This work is certainly worth to be published.

Your overall scientific approach is quite interesting and sufficiently sound. The developed method applying HFACS and combining it with your structural equation numerical modeling is promising and you generated a number of interesting results from a huge amount of data.

Of course, I very much want to encourage you to continue with your work! For the manuscript I recommend only very minor revision, including consideration of modification of some wording. An issue that I missed was a statement considering potential effects of varying quality of input data. Accident databases providing manifold data unfortunately not only measured but subjective especially in relation to the assessment of human failures and the question of how consistent is a data base remains a critical issue. However, this is rather a more general comment than a request for a substantial revision. It however, can be addressed by a comment on your outlook on future research to maybe also included consideration of such aspects as well.

Overall, I am satisfied but was a little bit missing discussions on relations to SAFETY I and SAFETY II approaches of HOLLNAGEL and earlier applications of HFACS. Together with others he provided a quite comprehensive survey on developments in the field of maritime accidents, which would very well fit to the list of references:

Schröder-Hinrichs, JU., Hollnagel, E. & Baldauf, M. (2012) From Titanic to Costa Concordia: A century of lessons not learned. WMU J Marit Affairs (2012) 11: 151. https://doi.org/10.1007/s13437-012-0032-3

Once again, overall the way you conduct and present your work complies with the requirements of scientific writing and publishing, only, if at all, minor revisions are needed!

I wish you good luck and much of success for your further work.

Kind regards

Author Response

first of all, We would also like to acknowledge the insight contributions from the anonymous reviewers whose thoughtful comments have helped to improve an earlier version of this paper.

About the chapter II, we updated more detail. In particular, we have added five articles on the status quo and progress of theoretical research on the cause of accidents. Emphasis is placed on the development of the understanding of the cause of accidents after the introduction of the complexity system. Safety I and Safety II principals were analysized. based on safety-I, we put forward to study the path dependence from complexity theory, which has the non-linearity and the impact of protective structures. 

Chapter 3, about table 2, we add the description: 

Here the original framework and structure proposed by Shappell (1997) was reserved, such as SL1 (X13, X14, X15, X16), SL2 (X8, X9, X10, X11, X12),SL3 (X4, X5, X6, X7), and SL4 (X1, X2, X3).

and

All marine accidents will be affected and controlled by human factors, ship factors, environmental factors and management factors. However, the manifestations of system factors vary greatly in different accidents. In order to assist in the implementation of accident case analysis, an accident analysis system needs to be designed to fully define the description and characterization of the cause of the accident. This step relies on historical data and subject-matter experts analysis from the latent sources, such as databases, experiments, simulations, webs and logical analytical models.

Chapter 4, about SEM, we add the description: 

The Structural Equation Model(SEM) is a statistical method that analyzes the relationship among different variables by using a covariance matrix of variables. The structural equation model integrates path analysis, confirmatory factor analysis and general statistical test methods to analyze the causal relationship between variables, including the advantages of factor analysis and path analysis. At the same time, it makes up for the shortcomings of factor analysis, taking into account the error factors, and does not need to be limited by the assumptions of path analysis. Based on this, we propose the strong and weak associated path of accident cause to quantitatively describe the mechanism of the accident.

In chapter 5, about the data,we add the description: 5.1.2

As to HFACS, human factors are those factors related to people in the operation of the system. Human factors are beneficial to the safety side (such as people exert their own ingenuity, overcome the adverse effects of mechanical equipment or harsh environment, etc.), but also can have a negative effect. As a research object of human factors in MTA, the negative impact on human safety due to human factors, namely Human Error was more important. The detailed information about the observed characters in accident reports were structured and formatted(also shown in Table 2).

Each sample analysis for the causes of the accident is based on observed characters items, such as management software , ship (cargo) hardware, environment(including natural conditions, geographical conditions, traffic conditions), and liveware (Xi et al., 2017). 

In the case of structured accidents’ documents, the observed characters in causes of the accidents are divided into the following 7 categories.

In chapter 5, about the data analysis,we add the description: 

First, quantitative data assignment is used for the extent of each factor's effect. According to the Level of impact, the rating is separately defined. Such as no effect, 5; involved, 4; mainly, 2; directly, 1. As to how the accident is described , for example, those which are described as a general accident, the detail influence factors which result to a certain accident includes observed characters such as “non finding in operation arrangements or process issues”, “Insufficient staff training time” and “VTS monitoring failure” (variable in Table 2). These factors effect the accident at different levels of influence as discussed above, namely, “directly”,” involved” and “no effect” respectively. That means the score is 1, 4, 5 respectively.Each accident sample can be described by the influence factors. 

Second,accident causal factors depends on the minimum score among the corresponding observed characters collected. As to the case statemented above, those 3 observed characters involved “Inadequate Supervision” were numerical analyse, and the lowest score is measured as 1, which means “directly”. Therefore, x4“Inadequate Supervision” is measured as 1. All the structured observed characters in accident reports were formatted to numerical analysis data. The tested data statistics are shown in Table 4.

In chapter 5, about the case study,we amend the description: 

5.4 Reliability analysis in path dependency

5.5 Sensitivity analysis of HFACS-MTA based on SEM model 

Chapters  7, we amend the description: 

Structural equation modeling is a powerful research tool in the field of safety sciences, but the establishment of related models relies on the knowledge of relevant scientific fields. The setting of the implicit variables of the structural equations of accident causation theory and the setting of the relationship between hidden variables have the theoretical knowledge base of the maritime field. The setting and measurement of the measured variables corresponding to the hidden variables also have their theoretical basis. The structural equation model is only a mathematical expression of the theoretical knowledge base of the relevant scientific field, and it provides a tool for us to study related safety sciences.

Round  2

Reviewer 1 Report

The changes made by the authors have indeed improved the paper.

As a minor request, I would still like to see a bit more discussion on what the research described in the paper tells about maritime accidents and their causes and whether previous studies from the maritime domain support or contradict these conclusions.

Author Response

Thanks again.

in the conclusion,we give the following comments:

(3) We have seen that in recent times, the theory of accident causation based on system theory has greatly changed and developed the understanding of traditional accidents forming. In particular, the characteristics of safety is seen as the emergence of systems, with safety issues as a matter of control. The cause of the accident is not only to describe the components in system structure, but also to explain the interaction and coupling between the causal factors. This paper believes that a certain mathematical algorithm is used to analyze the degree of cross-linking between factors, describe the process of action between factors, and then determine the path of accident formation. This is a quantitative demonstration of the cheese model, revealing the path dependence of management defects in the field of marine safety affecting human behavior.

We also see that, to study the safety problems of the complex marine traffic system, it constructs a theoretical model of a complex system and proposes an accident cause structural hypothesis. Appropriate algorithms for the theoretical human-machine-control model can be used to understand the safety performance of marine traffic systems under different parameters through mathematical analysis. Combined with big data ideas and intelligent prediction theory, it provides an important basis for risk pre-warning and accident prevention. This will be a problem that will require further research.

Reviewer 2 Report

Overall, well reflected the requirements. Please check only a few things as follows;

Line 105: Please change from Maritime accident to Marine accident.

Line 533: Please modify it to 5.1.1.

Line 772: If figure 6 does not make much sense, what about removing it?

Author Response

Thanks again. all have been amended.